# Oral intake of *Lactiplantibacillus pentosus* LPG1 Produces a Beneficial Regulation of Gut Microbiota in Healthy Persons: A Randomised, Placebo-Controlled, Single-Blind Trial

**DOI:** 10.3390/nu15081931

**Published:** 2023-04-17

**Authors:** Elio López-García, Antonio Benítez-Cabello, Antonio Pablo Arenas-de Larriva, Francisco Miguel Gutierrez-Mariscal, Pablo Pérez-Martínez, Elena María Yubero-Serrano, Antonio Garrido-Fernández, Francisco Noé Arroyo-López

**Affiliations:** 1Food Biotechnology Department. Instituto de la Grasa (CSIC), Carretera Utrera km 1, Campus Universitario Pablo de Olavide, Building 46, 41013 Seville, Spain; elopez@ig.csic.es (E.L.-G.); abenitez@ig.csic.es (A.B.-C.); agarrido@ig.csic.es (A.G.-F.);; 2Unidad de Gestión Clinica Medicina Interna, Lipids and Atherosclerosis Unit, Maimonides Institute for Biomedical Research in Córdoba, Reina Sofia University Hospital, University of Córdoba, 14004 Córdoba, Spain; aparenaslarriva@gmail.com (A.P.A.-d.L.); francisco.gutierrez@imibic.org (F.M.G.-M.); pablopermar@yahoo.es (P.P.-M.); 3CIBER Physiopathology of Obesity and Nutrition (CIBEROBN), Institute of Health Carlos III, 28029 Madrid, Spain

**Keywords:** compositional data analysis (CoDA), lactic acid bacteria, metataxonomic analysis, probiotic, table olives, vegetable fermenter

## Abstract

The search for vegetable-origin probiotic microorganisms is a recent area of interest. This study conducted a phase I clinical trial to assess the effects of oral administration of *Lactiplantibacillus pentosus* LPG1, a natural strain with probiotic potential isolated from table olive fermentations, on the gut microbiota. The trial was a randomised, placebo-controlled, single-blind study involving 39 healthy volunteers. Group A (*n* = 20) ingested one capsule/day of *L. pentosus* LPG1 containing 1 × 10^10^ UFC/capsule, while Group B (*n* = 19) received one capsule/day containing only dextrose (placebo). The capsules were taken during breakfast for 30 consecutive days. Human stool samples were collected from all volunteers at the beginning (baseline) and at the end of the study (post-intervention) and were subjected to 16S rRNA metataxonomic analysis using Illumina MiSeq. Sequencing data at the genus level were statistically analysed using traditional methods and compositional data analysis (CoDA). After treatment, the alpha diversity in Group B (placebo) decreased according to an increase in the Berger and Parker dominance index (*p*-value < 0.05); moreover, dominance D increased and Simpson 1-D index decreased (*p*-value < 0.10). The *Lactobacillus* genus in the faeces was included in the CoDA signature balances (*selbal* and *coda4microbiome*) and played a notable role in distinguishing samples from baseline and post-intervention in Group A (LPG1). Additionally, ingesting *L. pentosus* LPG1 modified the gut microbiota post-intervention, increasing the presence of *Parabacteroides* and *Agathobacter*, but reducing *Prevotella*. These findings suggest that *L. pentosus* LPG1 is a potentially beneficial gut microbiota modulator in healthy persons.

## 1. Introduction

Traditionally, fermented dairy products have dominated the probiotic food market. However, the demand for vegetal-based probiotic products is growing due to a shifting consumer preference for new alternatives, particularly among vegetarians and lactose-intolerant individuals. To meet this demand, the production of new vegetal-based probiotic foods should be promoted. In recent years, table olives have emerged as a promising alternative to dairy products for carrying beneficial microorganisms to consumers. *Lactiplantibacillus pentosus* (ex-*Lactobacillus pentosus*) strains can form biofilms on the olive epidermis, resulting in fermented products with high concentrations of microorganisms, reaching more than 10 million UFC/g [1]. 

*Lactiplantibacillus pentosus* LPG1 (from now on LPG1) is a ferment obtained from table olive biofilms with proven probiotic features in vitro and in vivo (murine model) studies. This strain has anti-inflammatory and phytase activities, can reduce cholesterol levels, inhibits food-borne pathogens, adheres to Caco-2 cells, and does not have haemolytic activity, among other features [2,3]. Moreover, clinical studies have confirmed its safe use in humans. In this sense, a recent study has revealed the *L. pentosus* LPG1 lacks antibiotic resistance and virulence genes in its genome but has many potential probiotics, bacteriocin, exopolysaccharides, folate production cluster genes, and enzymes capable of degrading complex carbohydrates such as galactose, glycogen, starch, cellulose, or xylan [4]. 

The probiotic potential of *Lactiplantibacillus* strains has been tested in previous clinical trials in humans. Kotani et al. (2010) investigated the ability of *L. pentosus* b240, initially isolated from fermented tea leaves, to accelerate salivary immunoglobulin A secretion in elderly individuals [5]. Wang et al. (2014) studied the effect of oral consumption of *Lactiplantibacillus plantarum* Lp-8 on the faecal microbiota composition [6]. Vos et al. (2017) examined the influence of oral administration of diverse *L. plantarum* strains on the immune response [7], while Rudzki et al. (2019) studied the impact of the oral intake of *L. plantarum* 299 on cognitive functions in patients with depression [8]. Recently, Ahn et al. (2020) studied the effects of an *L. pentosus* strain, isolated from kimchi, on children with allergen-sensitised atopic dermatitis [9]. Therefore, searching for *Lactiplantibacillus* strains with probiotic potential from vegetable sources is challenging. 

Data sets collected by high-throughput sequencing (HTS) of 16S rRNA gene amplimers, metagenomes, or meta-transcriptomes are common. They are widely used to investigate healthy and disease states in humans, food fermentations, and many other biological environments. Until now, the analysis of these data sets has mainly been based on multivariate statistical tools developed for unconstrained values and has yet to consider their specific properties. Gloor et al. (2017) alerted researchers about the dangers of ignoring the compositional nature of the HTS data sets derived from microbiome studies [10]. They pointed out that the total number of counts or read depth is a major confounder for distances or dissimilarity calculations for multivariate ordination. Rivera-Pinto (2018) described the issues derived from ignoring the microbiome data set compositionality (counts per sample constrained to the sequencing depth), as they do not represent the true microbiome abundance in the sample [11]. The main problems are spurious correlations, sub-compositional incoherence, and increased type I error [10,12,13]. Thus, the metataxonomic figures derived from this study were considered compositional data (CoDa) and were analysed accordingly. 

This study aimed to evaluate the influence of the oral administration of the potential probiotic from plant origin *L. pentosus* LPG1 on the bacterial gut microbiome in healthy volunteers using standard and CoDa analysis. 

## 2. Materials and Methods

### 2.1. Preparation of the Capsules 

White hypromellose type 0 capsules of plant origin for slow release (Nutra’V, Qualicaps Europe, Madrid, Spain), with a length of 10.32–11.12 mm, were selected for the clinical trial. Two formulations were provided by Bioges Starter S.A. (León, Spain, health certificate 31.02073/L.E.): (A) LPG1 capsules (*n* = 600) containing 0.46 g of product with 1 × 10^10^ UFC of lyophilised LPG1 + dextrose per capsule, and (B) placebo capsules (*n* = 600) containing only 0.46 g dextrose per capsule. The probiotic and placebo capsules were identical in colour, taste, smell, and packing. For distribution among volunteers, capsules were packaged in a rigid amber high-density polyethylene medical-grade bottle with an airtight seal (Labbox Labware, Barcelona, Spain). A bottle with 30 capsules (LPG1 or placebo) was supplied to each participant at the beginning of the study and kept until ingestion at 8 °C during the assay.

### 2.2. Clinical Trial

The essay consisted of a randomised, single-blind, single-centre, parallel pilot safety study with healthy individuals. The clinical trial was conducted at the Maimónides Biomedical Research Institute (IMIBIC, Cordoba, Spain) and the Reina Sofía University Hospital (Cordoba, Spain), where the screening, selection, and recruitment of the volunteers who participated in the study were carried out. The study was conducted following the Declaration of Helsinki, and the intervention protocol was approved by the Human Investigation Review Committee of the Reina Sofia University Hospital (number 2519-N-20) according to Institutional and Good Clinical Practice guidelines. All formulations were dispensed by a technician, with the investigator and subjects blinded to the identity of the intervention. Compliance was assessed from the weight of residual powder. All staff members involved in measuring any outcome were blind to the assignments. 

Healthy subjects were enrolled via public advertising. A total of 73 initial volunteers was contacted among those willing to enter the study and screened, involving a standardised and comprehensive medical history, physical examination, and clinical chemistry analysis before enrolment. Internists conducted the screening, selection, and recruitment processes between July and September 2021. Appendix A details the inclusion and exclusion criteria. All participants gave written informed consent before inclusion. Finally, 39 volunteers fulfilled the inclusion and exclusion criteria and participated in the study (October–November 2021). The sample size of participants was calculated based on previous findings, showing that oral administration of *Lactiplantibacillus* (in both capsules and fermented olives) could modify the intestinal microbiota [6,14]. 

Participants enrolled in the study were randomly assigned (1:1) by a computerised random sequence generator to orally ingest LPG1 capsules (one per day, Group A; here-forward Group A—LPG1) or placebo capsules (one per day, Group B; here-forward Group B—placebo). Capsules were swallowed during breakfast for 30 consecutive days. The participants had no standardised diet; however, ingesting antibiotics, antihistamines, or probiotics (supplements or certain foods such as yogurts, any type of olives, or fermented vegetables) during the study was strictly prohibited. 

### 2.3. Baseline Characterisation of Groups

Patients were given an appointment at 8.00 a.m., following a 12 h fast, and were admitted to the laboratory for anthropometric and biochemical tests. Anthropometric parameters were measured by trained dietitians using calibrated scales (BF511 Body Composition Analyzer/Scale, OMROM, Kioto, Japan) and a wall-mounted stadiometer (Seca 242, HealthCheck Systems, Brooklyn, NY, USA). Body mass index (BMI) was then calculated as weight per square meter (kg/m^2^). Systolic and diastolic blood pressure were measured with a validated digital automated blood pressure monitor.

Venous blood samples were collected from the antecubital vein in Vacutainer^TM^ tubes containing EDTA or no anticoagulant. Serum fasting glucose, total cholesterol, high-density lipoprotein cholesterol (HDL-C), and triglyceride (TG) levels were measured by spectrophotometry using an Architect c-16000 analyser (Abbot^®^, Chicago, IL, USA). Low-density lipoprotein cholesterol (LDL-C) was calculated using the Friedewald formula (provided serum TG levels were <400 mg/dL). Plasma fasting insulin levels were determined by chemiluminescent microparticle immunoassay using an Architect i-2000 analyser (Abbott^®^). Plasma concentrations of hsCRP were determined by a high-sensitivity ELISA (BioCheck, Inc., Foster City, CA, USA).

Groups A—LPG1 and B—placebo were composed of 20 (9 males and 11 females, with an average age of 31.4 years) and 19 persons (10 males and 9 females, with an average age of 33.6 years), respectively (Table 1). 

### 2.4. Human Faeces Processing

Bacterial DNA from faecal samples for each participant was isolated and purified using a ZymoBIOMICS^TM^ DNA/RNA Miniprep kit (Zymo Research, Irvine, CA, USA) according to the manufacturer’s instructions. Samples were collected before capsule administration (baseline) and after 30 days of oral intake (post-intervention). Thereby, a total of 78 faecal samples were processed. For this, an aliquot of around 250 mg of each faecal sample homogenised in saline solution (0.9% NaCl) in DNA/RNA Shield^TM^ faecal collection tubes (Zymo Research, Irvine, CA, USA) was processed. Purified DNA was then frozen at −20 °C until further metataxonomic analysis. Prior to sequencing, DNA concentration was measured using a Qubit 4 fluorometer (Thermo Fisher Scientific, Madrid, Spain), always reaching values above 5 ng/µL.

### 2.5. Metataxonomic Analysis

Massive sequencing was carried out at the FISABIO facilities (Valencia, Spain). For the bacterial populations, the V3 and V4 regions (459 bp) of the 16S ribosomal RNA gene were amplified with the designed primers surrounding conserved regions [15] following the Illumina amplicon libraries protocol. DNA amplicon libraries were generated using a limited PCR cycle: initial denaturation at 95 °C for 3 min, followed by 25 cycles of annealing (95 °C for 30 s, 55 °C for 30 s, 72 °C for 30 s), and a final extension at 72 °C for 5 min, using a KAPA HiFi HotStart ReadyMix (KK2602). Then, the Illumina sequencing adaptors and dual-index barcodes (Nextera XT index kit v2, FC-131-2001) were added to the amplicons. Libraries were normalised and pooled before sequencing. The pool containing indexed amplicons was loaded on the MiSeq reagent cartridge v3 (MS-102-3003) spiked with 25% PhiX control to improve base calling during sequencing, as recommended by Illumina for amplicon sequencing. Sequencing was conducted using a paired-end, 2 × 300 bp cycle run on an Illumina MiSeq sequencing system. Metataxonomics data were first analysed using the R package phyloseq 1.32.0 under default parameters. For each sample, Amplicon Sequence Variants (ASVs) were retained; the remaining reads were clustered against those ASVs allowing one mismatch to correct for error sequencing. Bacterial taxonomy was assigned using the SILVA 138 SSU database [16].

### 2.6. Analysis of Gut Bacterial Diversity

The alpha-biodiversity was studied using the Past statistical program [17] and the classical methodology, with richness (the number of different genera present) and evenness (homogeneity in the abundance of genera) as the main parameters. Observed richness, TAXA, is the most straightforward parameter to measure richness, which tends to underestimate the actual value. Chao 1 is the most extended index to compensate for the non-detected part (genus). Regarding evenness, the maximum value is observed when the abundance is uniformly distributed, and it is very low when only a few taxa account for most of the relative abundance in the sample. Information on other parameters developed to estimate alpha diversity can be found elsewhere [17]. Comparisons of diversity data among treatments (combinations of groups and essay phases: AI, AF, BI, BF) were analysed using the plugin XLSTAT v.2017 (Addinsoft, Paris, France).

### 2.7. CoDa Analysis of Gut Microbiota

The experimental matrix used for CoDa analysis, obtained from Appendix A, consisted of two sections. The first described the experimental conditions, followed by another section including the microbiome abundance of each volunteer. The overall matrix consisted of *n* rows, each associated with a particular volunteer, and *k* columns (environmental variables plus the taxa (ASVs) deduced from the metataxonomic analysis. Then, each row described the experimental conditions for each volunteer in the first section, followed by its gut microbiome composition of the corresponding volunteer in the second section. For the analysis, both sections were considered together or individually. Especially the gut microbiome abundance section deserves special attention. Each cell *x_ij_* represents the number of sequences (reads) corresponding to taxon *j* in sample *i*. The main characteristics of this microbiome abundance matrix are as follows: (i) the total number of counts among volunteers is highly variable, (ii) these counts are constrained by the maximum number of sequence reads of the sequencer, and (iii) the data contain a large proportion of zeros [10,12,18]. 

The first issue is usually addressed by normalisation before the analysis, but treating the data set as compositional does not require this pre-treatment. Regarding the second aspect, the total count constraint induces a strong dependence on the abundance of taxa (the increase in one taxon decreases the counts of others since their total number cannot exceed the sequencing depth specified). CoDa analysis adequately addresses this feature, minimising its effect in this work. The high proportion of zeros (sparsity) is considered zero counts and handled according to the package or program manuals. Usually, they are replaced by the values imputed using the Bayesian-Multiplicative (BM) treatment, as proposed by Palarea-Albaladejo and Martín-Fernández [19] or substituted with 1 in the abundance matrix. BM does not only suggest values for the zero cells but also modifies the non-zero values, although maintaining the original ratios between parts without zeros. Rivera-Pinto (2018) compared both methods and concluded that the results were somewhat similar [11]. 

Among the conditions required for a proper CoDa analysis are permutation invariance, scale invariance, and sub-compositional coherence, since one usually works with sub-compositions [10,12,18]. The CoDa packages used in the statistical analysis were *selbal* [11] and *coda4microbiome* [20]. R packages were run in R v 4.2.2 (R Core Team, 2022), under RStudio 2022.12.0+353, “Elsbeth Geranium” Release 2022-12-03 for Windows.

## 3. Results and Discussion

This work comprised a clinical phase I study with 39 healthy volunteers to determine the effects on the gut microbiota of oral administration of LPG1, a native ferment with probiotic potential isolated from table olive fermentations [2,3]. Baseline clinical and metabolic characteristics and lipid profiles of the participants according to groups (A-LPG1 and B-placebo) are presented in Table 1. No significant differences were observed between randomised groups in terms of baseline characteristics. Lyophilised LPG1 did not decrease in terms of population level (1 × 10^10^ UFC) in the capsules during the 30 days of the trial. The strain was confirmed to be a safe microorganism without significant differences from the placebo group after analysing diverse blood and urine biochemical parameters, anthropometric data, and expression of oxidative and anti-inflammatory markers. These data are unsurprising since the study was conducted with a group of healthy people. At the same time, the most relevant probiotic properties of LPG1 are related to its anti-inflammatory potential, proved in a murine model [2]. For this reason, it is planned to conduct a second clinical intervention trial with people suffering from allergic or immune diseases. However, this research has already found interesting effects on the human gut microbiota, as described below. 

### 3.1. Bacterial Diversity of Volunteers’ Faeces

After filtering and quality depuration, the 78 human stool samples analysed provided 2,703,556 sequences. The mean number of sequences obtained per sample was 34,661, ranging from a minimum of 24,132 to a maximum of 44,838. The metataxonomic analysis revealed 266 different ASVs bacterial genera (Appendix A. Before the statistical analysis, those genera with presence in only one subject or very low reads in two volunteers were removed, reducing the genera under statistical study to 121. From these, only 25 bacterial genera had a frequency of occurrence >1% in at least one of the samples (Figure 1). 

All the alpha diversity indices in the Past program [17] were evaluated for each group (A—LPG1 or B—placebo) according to the experiment phase (I, initial, or F, final), leading to four treatments (AI, AF, BI, and BF). The overall and specific values of the diverse parameters obtained (bootstrap = 1000) for the various treatments are shown in (Table 2). Five parameters remained unchanged during the essay period (TAXA_S, Menhinick, Margalef, Physer-alpha, and Chao-1), but the rest showed significant differences in at least one of the contrasts. 

The significant diversity parameters were studied in detail (Figure 2A–H). The individuals ranged from a minimum of 20,142 to a maximum of 45,067, and there was a slight decrease in the averages at the end of the experiment (Figure 2A), with the only significant difference observed being between AI and BF. Dominance D may oscillate from 0 (all taxa are equally present) to 1 (one taxon dominates the community completely). It ranged from 0.212 to 0.330, indicating a relatively similar taxa presence distribution, with the trend of averages slightly increasing after the trial in Groups A and B, although always showing scarce dominance. The significant differences were caused by the low value in BI (Figure 2B). Simpson index (1-D) is a measure of the “evenness” of the community and ranges from 0 (absolute lack of evenness) to 1 (complete evenness), with values close to 1 in the study indicating a somewhat homogeneous distribution of taxa. Due to its relationship with Dominance D, its trend was opposed to it, and the significant comparisons were the same (Figure 2C). The Shannon H index (entropy) is a diversity index that considers the number of individuals and the number of taxa and ranges from 0 for communities with only a single taxon to high values for those with many taxa, each with few individuals. In the assay, regardless of treatments, its values ranged from 2.089 to 3.712 (Table 2), indicating a moderate number of taxa, and the trend within groups slightly decreased after the trial (Figure 2D), with significant differences concerning AI and AF vs. BI because of the low and high values, respectively. Buzas and Gibson’s evenness (*e^H^/S,* with S being the number of taxa) (Figure 2E) is related to the Shannon H index. Its trend resembled Simpson 1-D and Shannon H. Brillouin (Figure 2F) and Equitability J (Figure 2G) also had similar trends (averages and significant differences) to those mentioned before. Berger and Parker’s dominance index is the number of individuals in the dominant taxon relative to *n*, the total number of individuals. It varies between 0 and 1 (the closer it is to 1, the greater the dominance and the lower the diversity). It changed as Individuals because of their close relationship. Its increase after treatment was significant only in Group B—placebo (Figure 2H). 

Our data partially agree with those obtained by Ahn et al. (2020), who found that gut microbiota diversity in children was similar in both placebo and probiotic groups after oral administration of an *L. pentosus* strain isolated from kimchi [9]. However, there are some differences between the two studies. In our case, the Berger and Parker dominance index significantly increased after the trial (*p* = 0.038, Figure 2H) in Group B—placebo. Additionally, we found a close to significant (*p* < 0.50) increase in Dominance D (*p* = 0.052, Figure 2B), and a decrease in Simpson 1-D (*p* = 0.054, Figure 2C) was also close to significant at *p* < 0.05 (significant at *p* < 0.10) in this group, which suggest a reduction in gut microbiota diversity. These results should be interpreted with caution. However, taken together with the other results of the study, the significant changes in these indexes in Group B—placebo suggests that oral intake of LPG1 may have a positive effect on gut microbial diversity. Human health is closely linked to the diverse set of intestinal microorganisms, collectively known as the gut microbiota [21]. While gut microbial diversity is decreasing, the prevalence of chronic inflammatory diseases such as inflammatory bowel disease, diabetes, obesity, allergies, and asthma is on the rise in Westernised societies [22]. A good balance of the gut microbiota is crucial for maintaining good health, and consuming probiotic products might help to resume and improve gut bacteria balance [23,24]. 

### 3.2. Predominant Bacterial Genera

The human intestine is populated by a large and dense microbial community that can be classified into three main groups or genera based on their relative abundance in the gut microbial community: *Bacteroides* (enterotype 1), *Prevotella* (enterotype 2), and *Ruminococcus* (enterotype 3) [25]. 

*Bacteroides* were the most frequently detected genus in all samples analysed, and their presence increased during the study in both groups (A—LPG1 and B—placebo) (Figure 1). The final highest levels (24.76%) of *Bacteroides* were found in Group A—LPG1, although the improvement during the assay was similar in both groups. *Bacteroides* predominant in enterotype 1 are usually associated with a diet high in protein and fat [25]. The presence of this bacterial genus in the human intestine is generally considered beneficial. It metabolises different oligosaccharides and polysaccharides and synthesises vitamins for the human body [26].

The second most relevant genus in abundance was *Faecalibacterium*, whose frequency also increased in both groups after the 30 days of the study (Figure 1) but was higher in Group B—placebo, reaching 14.16% at the end of the trial. The increase in *Faecalibacterium* presence in this group during the assay was statistically selected by CoDA software for segregating between the initial and final trial samples. The production of butyrate, short-chain fatty acids, and dietary fibre fermentation are among this bacterium’s beneficial properties. Its presence has been associated with an improved immune response [27], while low levels are associated with obesity, Crohn’s disease, and asthma [28].

The third genus in the percentage of occurrence was *Ruminococcus*. Its population levels decreased after the clinical trial in A—LPG1 and B—placebo groups (Figure 1). This genus is considered beneficial since it is involved in breaking plant cell walls and digesting complex carbohydrates, producing metabolites, short-chain fatty acids, and immune system maturation. Their populations decrease in inflammation and Parkinson’s disease [29].

The fourth important genus was *Prevotella*. Its frequency of appearance increased in the two groups after the 30 days of the clinical trial but mainly in the B—placebo group. In the A—LPG1 group, its levels did not exceed 3.94% at the end of the trial, while in the B—placebo group they reached 11.11%. As described below, this increase of *Prevotella* counts in Group B—placebo at the end of the study was statistically selected by CoDa analysis. A carbohydrate-rich diet could favour this bacterial genus’s development, which can also help break down polysaccharides. However, an excessive increase of *Prevotella* is associated with inflammation of the colon and a reduction in the production of short-chain fatty acids, being a potential pathobiont [30,31]. 

### 3.3. Metataxonomic CoDa Analysis

Because of the compositional nature of HTS data sets, the results of microbiome studies should be analysed using CoDa tools [10]. We used various free CoDa software available on the web. These programs were initially designed for the medical sciences to distinguish between healthy patients and those affected by specific illnesses. The objective of the analysis was to develop signature relationships (balances) that could differentiate between the gut microbiota at the baseline (I) and post-intervention (F) phases within the A—LPG1 and B—placebo groups. 

#### 3.3.1. Applying selbal

The package selbal is a software that identifies microbial signatures by modelling the response variable and determining the smallest number of taxa with the highest prediction or classification accuracy [11]. To select the log ratios, the program retains the first log ratio of two ASVs that provides the highest accuracy. Then, a new ASV is incorporated in the numerator (N) or denominator (D), and it is kept if the new balance increases the prediction accuracy. Otherwise, it is discarded. 

After following these sequential steps with the Group A—LPG1 microbiome abundance matrix, several ASVs were chosen to be integrated into the numerator (N) or denominator (D), as summarised in Table 3. An accuracy of about 86% was achieved with only the logratio *Granulicatella/Lactobacillus*. Later, *Ruminiclostridium_5*, *Ruminococcus_1*, *Coprococcus_3*, and *Anaerostipes* (in N) and *Faecalicoccus*, *Ruminococcaceae_UCG-004,* and *Parabacteroides* (in D) were incorporated into the balance. The resulting new balance (signature) was then able to predict with 100% accuracy the samples from the initial and final trial steps within Group A—LPG1. When the same procedure was applied to Group B—placebo, the most appropriate first balance was the logratio *Granulicatella/Faecalibacterium* with 81.10% accuracy (Table 3). *Subdoligranulum, Angilakisella*, and *Fusicatenibacter* were successfully added to the numerator, while only *Faecalicoccus* was incorporated into the denominator. However, the resulting overall balance was not wholly successful (98.34%) in assigning the B—placebo samples to the corresponding trial phases. Furthermore, the accuracy could not be further improved by incorporating any other ASV.

Comparing the median and distribution of the balance scores obtained after applying *selbal* software is challenging. Regarding Group A—LPG1 (Figure 3), the median of the balance was lower at the end of the trial (F) than at the beginning of it (I). Notice that this decrease could be related to the selection of *Lactobacillus* for the denominator just in the first step. Its population increase by ingesting the probiotic capsules could, in turn, have decreased the signature balance at the end of the trial in A—LPG1. Moreover, the distribution curves of Groups A and B scores showed only relatively low overlap. In fact, the balance was able to reach 100% segregating accuracy. 

Conversely, *Lactobacillus* was always ignored in the selection to distinguish between the initial and final microbiome in samples from Group B—placebo. Only a few more ASVs could improve the first balance segregation power. Moreover, the definitive best one could not wholly differentiate between the microbiome composition of the two phases in this group. The important overlap of the distribution curves of the balance at both moments and the smaller distances between the medians in this group than in A—LPG1 could explain the lower segregation power in B—placebo. In summary, the study by *selbal* indicates that *Lactobacillus* contributes to differentiating between the gut microbiota composition at the beginning and final of the trial in Group A—LPG1, possibly due to the contribution of the ingested LPG1 capsules, but it does not play any role in the same task in B—placebo. 

According to Figure 1, the presence of *Lactobacillus* sequences was low in all samples analysed, not being one of the predominant genera detected in the volunteers’ gut microbiota (Figure 1). The mean frequency of *Lactobacillus* sequences increased in Group A—LPG1 from 0.02% (just before starting the study) to 0.07% (after 30 days of LPG1 administration). However, more relevant was the analysis of the number of volunteers in which the presence of *Lactobacillus* sequences was detected, increasing from 6 to 13 persons in Group A—LPG1 (35.0% increase) but decreasing from 9 to 3 persons in Group B—placebo (31.6% decrease). Daily oral intake of LPG1 capsules increased the number of persons with *Lactobacillus* in post-intervention faeces (13/20 in Group A—LPG1, only 3/19 in Group B—placebo). In a previous study, we found that LPG1 exhibited a similar in vitro ability to adhere to the human colon cell line (0.75%) than other *Lactobacillus* species used as probiotic controls [2]. Furthermore, in silico analysis of the LPG1 genome revealed the presence of genes associated with bile salt tolerance, adhesion, and gut persistence [4], which may explain its detection in the faeces of the A—LPG1 group. According to Walter (2008), only a few *Lactobacillus* species are genuine inhabitants of the mammalian intestinal tract, and most lactobacilli are allochthonous members derived from fermented food, the oral cavity, or more proximal parts of the gastrointestinal tract. Accardi et al. (2016) reported that an oral intake of 25 g of fermented olives for 30 days also increased the presence of *Lactobacilli* in faeces [14]. 

#### 3.3.2. Applying *coda4microbiome*

*coda4microbiome* is a recently developed R package to provide exploratory, cross-sectional, and longitudinal tools for analysing microbiome data within the CoDa framework [20]. The package aims to identify microbial signatures by selecting variables based on generalised linear (quantitative variables) or logistic (qualitative) models. One interesting feature of the exploratory section of *coda4microbiome* is the ability to visualise the log ratios of the ASVs that are more associated with the response (Y) through a heat map-like plot that includes the log ratios of the taxa with the highest prediction accuracy.

Both *selbal* and *coda4microbiome* have in common the search for two groups of taxa (numerator, N, and denominator, D), which, jointly, are highly associated with the response (Y, with levels I and F in this case). However, they differ in several aspects: In the model for combining the relative abundance of taxa in N and D, *selbal* uses a simple *ilr* (isometric log ratio transformation) balance (log of the geometric mean of taxa in N/geometric mean of taxa in D). In contrast, *coda4microbiome* uses log-contrasts of taxa (linear model), with the constraint that the sum of coefficients should sum 0. Those taxa with positive coefficients are assigned to N, while those with negative coefficients are assigned to D; the rest do not participate in the microbial signature. The two packages are also different regarding the variable selection algorithm: forward selection (*selbal*) and penalised regression (*coda4microbiome*) [11,20].

For exploratory analysis, *coda4microbiome* determines the association of each pairwise log ratio with a dependent variable. The information is interesting because an ASV highly associated with the response (Y) is likely to be associated with it regardless of the second taxa. The function *explore_log ratios* in *coda4microbiome* provides the importance of each ASV. Its contribution is evaluated through the prediction accuracy of the whole sets of log ratios. The most relevant results are shown as a colour-scaled heat map. In Group A—LPG1 (Figure 4, left), the log ratios of *Lactobacillus* vs. any other ASV were generally strongly associated (high prediction accuracy) with the response: the more intense the relationship, the darker the blue tone. The highest accuracy was obtained when associating *Lactobacillus* (in the D) with *Granulicatella*, *Streptococcus*, *Ruminiclostridium_5*, *Parabacteroides*, *Romboutsia*, *Anaerostipes*, *Ruminococaceae_UCG.013*, or *Actinomyces* (in D). These relationships agree with selecting by *selbal* the log ratio *Lactobacillus/Granulicatella* in the first step. The high accuracy observed to predict initial or post-treatment levels in Group A—LPG1 could be another clue to demonstrate that ingesting the probiotic LPG1 capsules had a positive response, and *Lactobacillus* may be an active factor in the modulation of the human gut microbiome. Overall, these findings suggest that *Lactobacillus* could be a potential therapeutic target for modulating the gut microbiome and improving human health.

On the contrary, the exploratory analysis of Group B—placebo data (Figure 4, right) did not reveal any relevant log ratio involving *Lactobacillus*. Instead, *Granulicatella* had a small role in this group. Additionally, log ratios including *Methanobrevibacter* vs. *Suttterella*, *Butyricimonas*, *Faecalicoccus*, *Parabactereroides*, or *Faecalibacterium* were found to be relevant in this group. Notice that this result is in agreement with the results obtained from applying *selbal*. The absence of *Lactobacillus* in associating the gut microbiome in samples with the initial and final trial phases agrees with the ingestion of only a placebo by these volunteers.

The cross-sectional analysis using *coda4microbiome* resulted in the selection of seven ASVs for the A—LPG1 group and ten ASVs for the B—placebo group, as shown in Figure 5. By design, the coefficients of the selected ASVs sum up to zero, indicating that the model is a log-contrast function. Similar to *selbal*, the balance scores obtained for the A—LPG1 group were lower in median and distribution than those obtained for the B—placebo group.

Note the selection of *Lactobacillus* with a relevant coefficient value in the A—LPG1 balance. This indicates that this genus plays a notable role in distinguishing samples from the initial and final trial phase in the A—LPG1 group. The negative sign assigned to *Lactobacillus* in the balance (in D) indicates that when its presence increases, the scores decrease, favouring the identification of samples collected at the end of the essay. Moreover, the high negative coefficient of *Parabacteroides* (in D) also represents a relevant contribution to identifying A—LPG1 final trial samples and agrees with its large increase in this group during the assay. Conversely, high values would be linked to samples (gut microbiota) collected at the initial step and associated with *Ruminiclostridium_5*, *Anaerostipes*, *Granulicatella*, and *Streptococus*.

The ASVs selected (Figure 5) for the balance to distinguish between the initial and final steps of the assay in Group B—placebo were different, except for *Granulicatella*. A decrease in those ASVs in the numerator (*Subdoligranulum*, *Granulicatella*, *Methanobrevibacter*, *Actinomyces*, *Terrisporobacter*, and *Dorea*) or an increase in those in the denominator (*Prevotella, Faecalibacterium, Butyricimonas,* and *Faecalicoccus*) tended to assign the samples from Group B—placebo to the final phase of the essay. However, notice that, again, *Lactobacillus* was absent from this balance and, subsequently, played an irrelevant role in assigning samples to the initial or final phases of the experiment. Overall, these findings suggest that *Lactobacillus* could be a potential therapeutic target for modulating the gut microbiome and improving human health.

According to Figure 1, the mean frequency of *Parabacteroides* sequences increased in Group A—LPG1 from 1.33% (just before starting the study) to 2.31% (after 30 days of LPG1 administration). In contrast, this increase was lower in Group B—placebo (from initial 1.44 to final 1.55%). This bacterial genus metabolises carbohydrates and produces short-chain fatty acids. *Parabacteroides* are currently being treated as a potential probiotic due to their anti-inflammatory effects and protection against obesity [32]. It was also reported that ingestion of *Lactobacillus* spp. produced an increase in *Parabacteroides* [33]. According to the CoDa analysis conducted in this work, daily oral intake of LPG1 capsules increased the number of *Parabacteroides* sequences in volunteers’ faeces (Figure 1) and, subsequently, could have a favourable health effect.

Although *Agathobacter* was not statistically selected during the CoDa analysis, there was a notable variation in the frequency of appearance of this genus during the clinical trial. The mean frequency of *Agathobacter* sequences increased in Group A—LPG1 from 4.67% (before starting the trial) to 4.98% (after 30 days of LPG1 administration). On the other hand, this genus decreased in Group B—placebo (from initial 3.70 to final 1.98%) (Figure 1). The bacteria of this genus are beneficial because they produce butyrate, and a decrease in its population levels is associated with sleep disorders in children [34]. On the contrary, an increase in its population levels due to the intake of prebiotics is associated with a reduction in cholesterol [33].

Diverse strains of *Lactiplantibacillus* have been shown to have probiotic effects in human clinical trials, such as *L. pentosus* b240, a strain of vegetable origin, which accelerated salivary immunoglobulin A secretion in older adults [5]. Wang et al. (2014) reported that oral intake of *L. plantarum* Lp-8 increased beneficial bacteria in faecal microbiota while decreasing opportunistic pathogens [6]. Vos et al. (2017) noticed that the impact of oral consumption of *L. plantarum* on host immunity is strain dependent, with some strains enhancing specific responses against pathogens [7]. Oral administration of *L. plantarum* Lp299 improved cognitive functions in patients with major depression by decreasing kynurenine concentration [8] and showed a systematic and local reduction of inflammatory response in healthy subjects [35]. Recently, Ahn et al. (2020) showed that an *L. pentosus* strain isolated from kimchi improved the treatment of children with allergen-sensitised atopic dermatitis [9]. All these studies show the enormous potential that certain *L. pentosus* and *L. plantarum* strains have for their use as human probiotics.

## 4. Conclusions

LPG1 has proved to be a potential probiotic microorganism because of its beneficial effects on the gut microbiota when administered orally. Using plant-based probiotics to modulate gut microbiota is a promising strategy for enhancing gastrointestinal health in humans.

## Figures and Tables

**Figure 1 nutrients-15-01931-f001:**
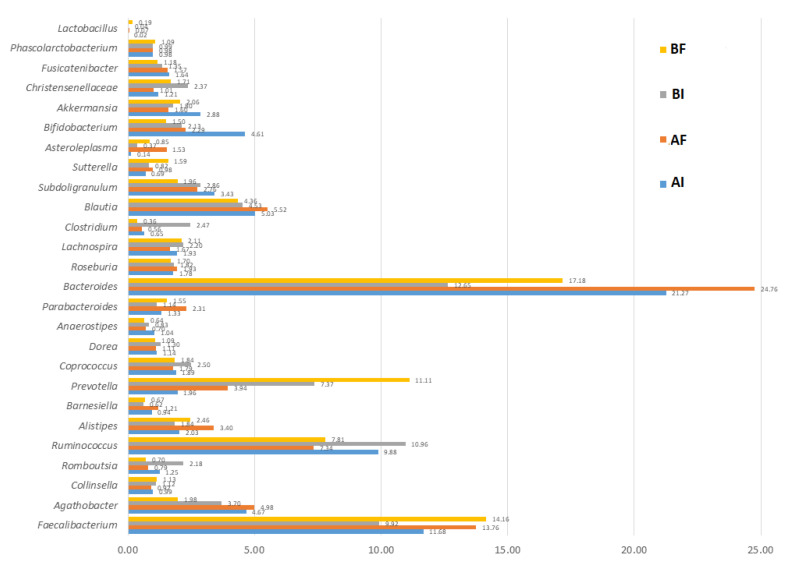
Majority bacterial genera found in the faeces samples of the randomised, placebo-controlled, single-blind trial to study the effects of oral intake of *L. pentosus* LPG1 on the human gut microbiota. The predominant gut microbial genera with a frequency of occurrence >1% in at least one faeces sample are plotted, according to treatments (AI, AF, BI, and BF for their respective groups and trial phases). Genus *Lactobacillus* (newly named *Lactiplantibacillus*) was also included, given its transcendence in the study. Regardless of the group and the assay phase, *Bacteroides*, *Ruminococcus*, and *Faecalibacterium* were the predominant genera. *Parabacteroides* and *Agathobacter* increased in the A—LPG1 group, while *Faeclibacterium* increased mainly in the B—placebo group.

**Figure 2 nutrients-15-01931-f002:**
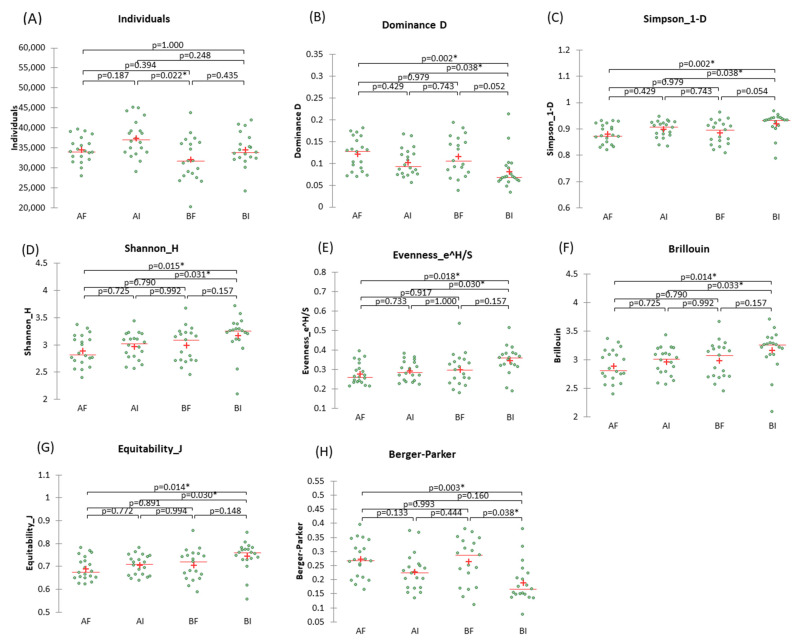
Dispersigrams of the alpha-diversity indices obtained in the randomised, placebo-controlled, single-blind trial to study the effect of oral *L. pentosus* LPG1 on the human gut microbiota. According to the Kruskal-Wallis test, at least one significant difference between treatments (AI, AF, BI, and BF for their respective groups and trial phases) was obtained. Notably, there are significant differences between the initial and final stages in the Group B—placebo, according to Berger-Parker (at *p* < 0.05) and Dominance D and Simpson 1-D indices (at *p* < 0.10). * stands for significant differences.

**Figure 3 nutrients-15-01931-f003:**
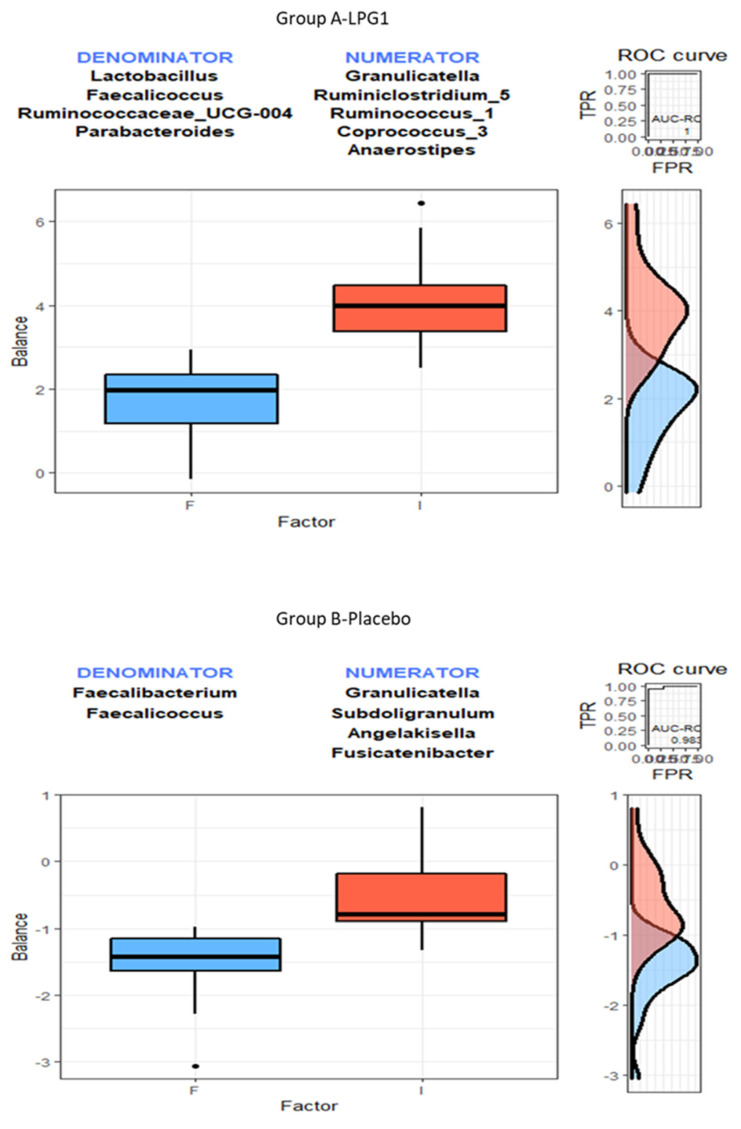
Boxplots and distribution curves of the balance scores selected by the *selbal* package (as shown in Table 3) generated from the gut microbiome data for each group included in the randomised, placebo-controlled, single-blind trial. The balance scores allowed for distinguishing between samples from the initial (I) and final (F) phases in A—LPG1 but were less effective for the B—placebo group (ROC values). Notably, the *Lactobacillus* genus was included in the balance for segregating the initial and final phases in the A—LPG1 group but was absent from the balance for the B—placebo group.

**Figure 4 nutrients-15-01931-f004:**
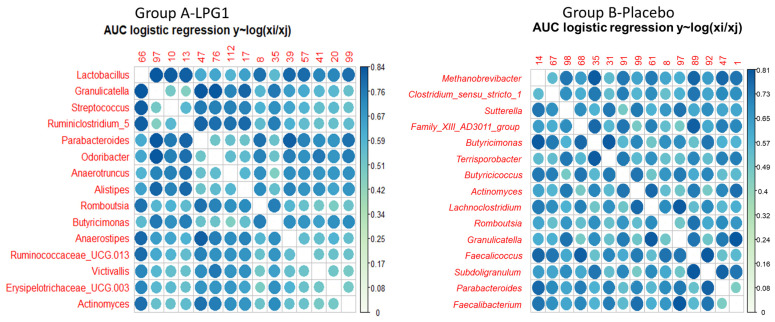
Heat-map-like plot (colour scale on the right) generated from the exploratory analysis of the *coda4microbiome* R package for the data obtained in the randomised, placebo-controlled, single-blind trial to study the effect of oral intake of *L. pentosus* LPG1 on the human gut microbiota. The plot displays the most relevant log ratios between ASVs from gut microbiota associated with the response (initial and final trial phases) in A—LPG1 and B—placebo groups. It is worth noticing that the log ratios *Lactobacillus* vs. *Granulicatella*, *Streptococcus*, *Anaerostipes*, *Ruminococaceae* UCG-013, or *Actinomyces* exhibited a high predictive accuracy in A—LPG1, while *Lactobacillus* did not play any role in B—placebo.

**Figure 5 nutrients-15-01931-f005:**
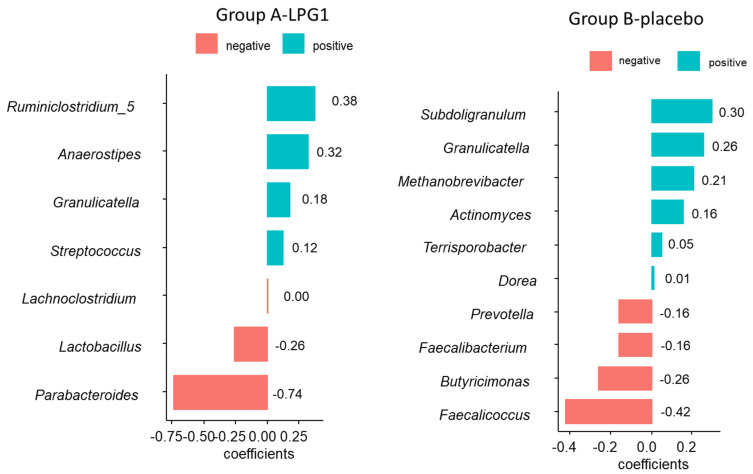
ASVs selected by *coda4microbiome* R package (logistic regression) for segregating samples from the initial (I) and final (F) phases of A—LPG1 and B—placebo groups included in the randomised, placebo-controlled, single-blind trial to study the effect of oral intake of *L. pentosus* LPG1 on the human gut microbiota.. The balance consisted of the positive coefficients in the numerator and the negative in the denominator. Notice that the *Lactobacillus* genus was selected to differentiate between trial phases in A—LPG1. The selection and increase in *Parabacteroides* during the experiment were also relevant to this aim. However, *Lactobacillus* did not play any role in segregating samples from the two B—placebo phases.

**Table 1 nutrients-15-01931-t001:** Baseline information of the groups included in the randomised, placebo-controlled, single-blind trial to study the effect of oral intake of *L. pentosus* LPG1 on the human gut microbiota. The data includes participants’ characterisation, the average baseline for their clinical parameters, metabolic and lipid profiles of the volunteers, and a statistical comparison between A-LPG1 and B-placebo groups. The *p*-values did not show any significant differences between groups.

Parameter	Group A—LPG1	Group B—Placebo	*p*-Value *
Number of participants	*n* = 20	*n* = 19	
Men/Women	9/11	10/9	
Age, years	31.45 ± 8.28	33.63 ± 6.96	0.380
Weight, kg	68.07 ± 13.13	70.97 ± 12.50	0.484
BMI, kg/m^2 #^	23.95 ± 2.50	24.82 ± 2.95	0.323
Total fat, %	30.60 ± 6.91	29.14 ± 9.79	0.601
SBP, mm Hg	117.73 ± 9.91	120. 66 ± 14.56	0.477
DBP, mm Hg	71.25 ± 9.55	71.83 ± 12.21	0.870
Fasting glucose, mg/dL	80.20 ± 6.11	78.88 ± 6.14	0.515
Fasting insulin, mU/L	7.85 ± 3.20	7.08 ± 3.72	0.495
Total cholesterol, mg/dL	174.55 ± 34.07	169.47 ± 26.77	0.609
LDL-cholesterol, mg/dL	104.35 ± 27.06	95.26 ± 25.67	0.290
HDL-cholesterol, mg/dL	53.05 ± 11.49	55.89 ± 16.08	0.527
Triglycerides, mg/dL	85.75 ± 41.30	91.31 ± 48.31	0.701
hsCRP, mg/dL	3.01 ± 5.13	1.15 ± 1.39	0.136

Unless otherwise stated, values are represented as the mean ± standard error or percentage of participants. * *p*-value for ANOVA analysis comparing Group A—LPG1 vs. Group B—placebo. **^#^** Body mass index (BMI) was calculated as weight in kg divided by the square of height in m (kg/m^2^). SBP, systolic blood pressure; DBP, diastolic blood pressure; hsCRP, highly sensitive C-reactive protein.

**Table 2 nutrients-15-01931-t002:** Alpha diversity indices obtained in the randomised, placebo-controlled, single-blind trial to study the effect of oral intake of *L. pentosus* LPG1 on the human gut microbiota. The table shows their minima, maxima, means, and standard deviations. Besides, it includes the *p*-value of comparing the values for such indices according to treatments (AI, AF, BI, and BF for their respective groups and trial phases) using the Kruskal-Wallis test. Significant (*p* < 0.05) differences between at least one of the contrasts between treatments are indicated in bold.

Variable	Minimum	Maximum	Mean	SE	K-W *p*-Value
TAXA_S	43.000	83.000	68.679	8.307	0.370
Individuals	20,142.000	45,067.000	34,660.974	4752.616	0.017
Dominance D	0.033	0.212	0.105	0.041	0.002
Simpson_1-D	0.788	0.967	0.895	0.041	0.002
Shannon_H	2.089	3.712	3.003	0.314	0.010
Evenness_*e^H^/S*	0.178	0.535	0.303	0.071	0.012
Brillouin	2.084	3.701	2.997	0.313	0.010
Menhinick	0.240	0.515	0.372	0.052	0.146
Margalef	4.048	7.848	6.481	0.797	0.353
Equitability_J	0.555	0.854	0.710	0.060	0.010
Fisher_alpha	4.893	10.310	8.253	1.132	0.371
Berger–Parker	0.075	0.394	0.238	0.077	0.002
Chao-1	43.000	83.000	68.692	8.328	0.372

Notes: Observations, *n* = 78; SE, standard error.

**Table 3 nutrients-15-01931-t003:** Results of applying the *selbal R* package to the metataxonomic data obtained in the randomised, placebo-controlled, single-blind trial to study the effect of oral intake of *L. pentosus* LPG1 on the human gut microbiota. The table shows differentiation between the human gut microbiota composition at the initial (I) and final phases of the trial in the A-LPG1 and B-placebo groups, using a CoDA balance (signature). The table includes the ASVs selected for the balance, their assignation to the numerator or denominator, and the accuracy and its increment after each selection step.

Step Order	Numerator	Denominator	Accuracy	Increase
**Group A—LPG1**				
1	*Granulicatella*	*Lactobacillus*	0.8588	0.8588
2	*Ruminiclostridium_5*		0.8850	0.0263
3		*Faecalicoccus*	0.9300	0.0450
4	*Ruminococcus_1*		0.9400	0.0100
5		*Ruminococcaceae_UCG-004*	0.9800	0.0400
6		*Parabacteroides*	0.9850	0.0050
7	*Coprococcus_3*		0.9975	0.0125
8	*Anaerostipes*		1.0000	0.0025
**Group B—placebo**				
1	*Granulicatella*	*Faecalibacterium*	0.8310	0.8310
2	*Subdoligranulum*		0.89474	0.06371
3		*Faecalicoccus*	0.94183	0.0360
4	*Angelakisella*		0.96399	0.02216
5	*Fusicatenibacter*		0.98338	0.01939

## Data Availability

The data and biological samples supporting this study’s findings are available from the corresponding author upon reasonable request to favour new collaborations.

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
