# Peer review of "Oral intake of Lactiplantibacillus pentosus LPG1 Produces a Beneficial Regulation of Gut Microbiota in Healthy Persons: A Randomised, Placebo-Controlled, Single-Blind Trial"

_nutrients, 2023, doi:10.3390/nu15081931_

Round 1

Reviewer 1 Report

The work presented shows the results of supplementation with lactobacilli in healthy people, which is quite interesting, since most of the studies are carried out with sick individuals.The model is well described, the study group well defined and the microbiota analyzes very well performed.I would like a graphical abstract of the experimental model and key results to be included.Captions for figures and tables could be improved to be more self-explanatory

Reviewer 2 Report

Manuscript submited by Elio," Oral intake of Lactiplantibacillus pentosus LPG1 produces a 2 beneficial regulation of gut microbiota in healthy persons: a 3 randomised, placebo-controlled, single-blind trial". 

Authors investigated the probiotic from plant source to check the its effect on the gut microbiota. The present study is poor decribed in terms of results and quality of English wirting. Authors have to resubmit this study by improving its quality of English and separating the  Resutls and discussion section. Methods should be concise and comprehensive. Further most of the results have same trend between placebo and treatment like for Figure 2A individuals diversity decreased both in BF and AF same for dominance, placebo resutls have minimum effects on BF. Figure A Bateroides is the most found genera but BF also shows same increasing trend as AF low value is due to BI low value. There are other concerns like mentioned above. Authors have to adress the concerns and can submit improved manuscript 

Reviewer 3 Report

Research well planned and properly executed. However, several aspects would need to be better described to demonstrate the beneficial effects on the intestinal microflora:

1. what metabolites does Lactiplantibacillus pentosus LPG1 produce?

2. does this strain have the ability to adhere to the intestinal epithelium?

3. are the small changes in the composition of the intestinal microflora presented not coincidental? E.g. change in diet.

4. what carbon sources does Lactiplantibacillus pentosus LPG1 use?

5. why did the number of Bifidobacterium cells decrease in this study?

Round 2

Reviewer 2 Report

Author's response is statisfactory. So it can be published